# Carbon nanotubes allow capture of krypton, barium and lead for multichannel biological X-ray fluorescence imaging

Christopher J. Serpell[1,2], Reida N. Rutte[1], Kalotina Geraki[3], Elzbieta Pach[4], Markus Martincic[5], Magdalena Kierkowicz[5], Sonia De Munari[1], Kim Wals[1,6], Ritu Raj[1], Belén Ballesteros[4], Gerard Tobias[5], Daniel C. Anthony[6] & Benjamin G. Davis[1]

The desire to study biology *in situ* has been aided by many imaging techniques. Among these, X-ray fluorescence (XRF) mapping permits observation of elemental distributions in a multichannel manner. However, XRF imaging is underused, in part, because of the difficulty in interpreting maps without an underlying cellular 'blueprint'; this could be supplied using contrast agents. Carbon nanotubes (CNTs) can be filled with a wide range of inorganic materials, and thus can be used as 'contrast agents' if biologically absent elements are encapsulated. Here we show that sealed single-walled CNTs filled with lead, barium and even krypton can be produced, and externally decorated with peptides to provide affinity for sub-cellular targets. The agents are able to highlight specific organelles in multiplexed XRF mapping, and are, in principle, a general and versatile tool for this, and other modes of biological imaging.

[1] Chemistry Research Laboratory, Department of Chemistry, University of Oxford, Mansfield Road, Oxford OX1 3TA, UK. [2] School of Physical Sciences, Ingram Building, University of Kent, Canterbury, Kent CT2 7NH, UK. [3] Diamond Light Source, Harwell Science and Innovation Campus, Didcot OX11 0DE, UK. [4] Catalan Institute of Nanoscience and Nanotechnology (ICN2), CSIC and The Barcelona Institute of Science and Technology, Campus UAB, Bellaterra, 08193 Barcelona, Spain. [5] Institut de Ciència de Materials de Barcelona (ICMAB-CSIC), Campus de la UAB, Bellaterra, 08193 Barcelona, Spain. [6] Department of Pharmacology, University of Oxford, Mansfield Road, Oxford OX1 3QT, UK. Correspondence and requests for materials should be addressed to B.G.D. (email: Ben.Davis@chem.ox.ac.uk).

Many imaging techniques that exploit electromagnetic radiation have been developed to probe and manipulate biological systems *in situ*. While certain regions of frequency have found widespread intensive application (Supplementary Fig. 1), such as hard X-rays (radiography, 3–30 EHz), radio waves (MRI, 3 KHz–300 MHz), UV–visible light (fluorescence microscopy, 430 THz–30 PHz) and gamma rays (SPECT/PET, >30 EHz), some potential uniquely powerful areas of the electromagnetic spectrum are notably underemployed. X-ray fluorescence (XRF) results from energy released when outer sphere electrons fall into holes in core atomic orbitals caused by ionization by the lower-energy end of hard X-ray radiation (500 PHz–5 EHz, equivalent to energies of 2–20 keV)[1]. The resulting sharp, characteristic emission lines in unoccupied regions suggest quantitative, ultra-multi-channel imaging (>20) using a single, sharp excitation wavelength, well beyond the limited multi-modal strategies of current biological imaging. Advantageously, conventional XRF imaging is not sensitive to the oxidation state and bonding of the imaged chemical element. However, full use of this prospective unique precision and high informational content necessitates use of further elements that are either gaseous or toxic as 'labels'. Here we show that the orthogonal filling and decoration of carbon nanotubes (CNTs) allows a versatile, modular system for the effective targeting of such intractable XRF contrast agent elements. These hermetically sealed CNT 'nanobottles' bearing organelle-specific peptides[2,3] allowed not only systematic delineation of key sub-cellular structures (cell membrane, nucleus and endoplasmic reticulum), using toxic barium, lead or even gaseous krypton as 'labels', but also allowed vital insight into the versatility of CNTs as carriers that are trafficked inside cells. Given their robust control of contents, large filling capacity and ready modular functionalization, we envisage that such CNTs will prove to be general, targetable, encapsulating imaging agents for this and other novel-mode imaging techniques.

## Results

**Design of XRF nanobottle contrast agents.** XRF elemental contrast agents were chosen that would not only allow distinct identification of emission peaks, but that would test the limits of both the filling capacity and proper hermetic sealing of any associated carrier system. Three different CNT filling 'materials'/elements were selected—Ba, Pb and Kr—determined by critical factors relating to their XRF properties.

Vitally, elements with emission energies less than ~2 keV are typically not observed due to absorption by air between emitting sample and detector, whereas those requiring excitation energies >18 keV receive a reduced flux of photons (and hence emit less) due to the spectral properties of radiation from typical sources, including even synchrotron beamlines[4], optimized for broad-range X-ray spectroscopy[5]. Furthermore, intervening emission regions of 2.0–3.7 and 5.9–8.6 keV are heavily populated by biologically abundant elemental XRF emissions that would therefore provide a high-potential background signal in any imaging applications (Supplementary Fig. 2). Barium ($L\alpha_1 = 4.5$ keV), lead ($L\alpha_1 = 10.5$ keV) and krypton ($K\alpha_1 = 12.6$ keV) occupy near ideal positions in the clear spectral 'windows' between these cluttered and/or low utility regions. Yet, their global toxicity (Pb), neurotoxicity (Ba) and gaseous state (Kr) would ordinarily prohibit their use. Their targeted use in isolation, or even in conventional conjugates as compounds, would therefore otherwise be extremely challenging to envisage in biological applications (either due to their toxic reactivity or essential lack of

reactivity) without due sequestration. While we have chosen three fillings here as illustrative examples, in principle any element or combination of elements may be contained with CNTs, providing the ability to produce highly multiplexed, 'barcoded' contrast agents.

We considered that proper capture and redirection of these elements might be uniquely achieved with CNTs for three reasons. First, CNTs have an exceptionally spacious cavity that can be filled with a wide variety of non-natural inorganic compounds (potential XRF tag elements) either through capillary force-induced uptake in the melt[6] or, potentially, gaseous diffusion[7,8]. Second, formation of impermeable hemispherical caps from open CNTs, for example, upon post-melt cooling[9], creates totally isolated microphases within sealed CNTs through which nothing larger than a proton can traverse the graphene membrane[10,11]. As a result, any toxicity or other biological activity of potential XRF tag elements is neutralized, since escape is prevented[12]; simultaneously any loss, leaching or degradation of the encapsulated, active label is prevented. Third, the outer covalent functionalization of these CNTs could, in principle, be achieved by a number of methods[13], allowing ready and flexible attachment of putative targeting ligands and/or modulating groups to control both biological function and compatibility, as well as providing a platform for the testing of probe moieties. Here, peptidic ligands for cellular receptors and trafficking proteins provided a versatile class of targeting moieties[2,3] to test the potential localization of these 'nanobottles' in an organelle-specific manner to sub-cellular structures (cell membrane, nucleus and endoplasmic reticulum). Together, these allowed the design and creation of a modular, addressable system (Fig. 1), through which we were able to systematically probe the effects of surface label upon localization of CNTs in cellular compartments.

**Construction and characterization of nanobottles.** Steam-opened and -shortened[14] (ca. 400–450 nm) and purified[15] single-walled CNTs (SWCNTs), which are widely regarded as biocompatible[16], were first tested as containers for lead. Enclosed under vacuum (to prevent oxidation by air upon annealing) within sealed quartz ampoules, open-ended and empty SWCNTs, in the presence of lead oxide were heated to 100 °C above the melting temperature of the inorganic material (888 °C) to initiate molten-phase capillary filing[17]. After slow annealing to ambient temperature, the samples were thoroughly sonicated and washed with concentrated hydrochloric acid to remove unencapsulated, toxic material. Pleasingly, a range of methods (X-ray photoelectron spectroscopy (XPS; Supplementary Fig. 3), XRF, high-angle annular dark-field scanning transmission electron microscopy (HAADF-STEM) and thermogravimetric analysis (TGA)) confirmed the presence of successfully encapsulated lead; in particular, HAADF-STEM measurements showed well-packed, condensed phase lead content in all cases (Fig. 2a; Supplementary Fig. 4). Vitally, Raman spectroscopy (Supplementary Fig. 5) confirmed that the nanotubes were essentially intact, with only a slight increase in the magnitude of D-band resonances associated with defects[18]. TGA further confirmed this successful capture of PbO (~15.2%) (Supplementary Fig. 6). Most importantly, synchrotron XRF spectroscopy revealed a powerfully strong signal from the incarcerated materials (Fig. 2b) with the designated Pb emission lines, emerging well above even the strongest background signals in a clearly distinct spectral window. Capping was measured by further attempts to remove PbO by treatment with concentrated hydrochloric acid (Supplementary Methods). Inductively coupled plasma mass spectrometry (ICP-MS) revealed that 51% of the encapsulated lead could be

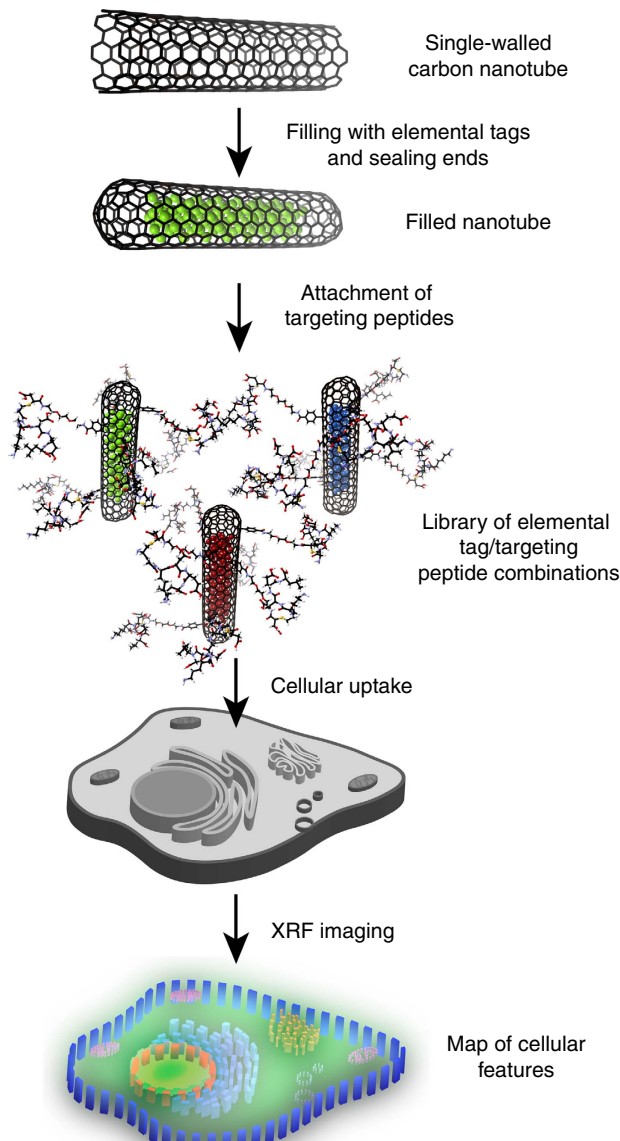

Single-walled carbon nanotube

Filling with elemental tags and sealing ends

Filled nanotube

Attachment of targeting peptides

Library of elemental tag/targeting peptide combinations

Cellular uptake

XRF imaging

Map of cellular features

**Figure 1 | Design of XRF-contrast carrier systems to probe cellular organelles through XRF mapping.** By filling, sealing and modifying CNTs, 'nanobottle' systems with targeting ligands can be created based upon almost any appropriate elemental system in a modular and general manner. Colour coding in this figure indicates generic multiplexed imaging rather than specific materials.

extracted in a first wash, but subsequent washes were lead-free (Supplementary Table 1). This indicated that although closure of the tube ends was not 100% (attributed to the oxidizing nature of the filling material), a significant level of Pb was irreversibly contained within doubly-capped SWCNTs nonetheless.

Having demonstrated successful encapsulation of lead, we chose next to fill CNTs with both barium and iodine to create a potential, 'dual-mode' XRF-active probe system bearing two independent sets of distinct emission lines in useful windows (Ba $L\alpha_1 = 4.5$ keV, I $L\alpha_1 = 3.9$ keV) as a proof of principle. Barium iodide (Ba $L\alpha_1 = 4.5$ keV, I $L\alpha_1 = 3.9$ keV) is a highly soluble salt that would normally be readily dispersed in the body, with iodide being consequently naturally concentrated in the thyroid[19], while barium ions are toxic, interfering with muscular and neuronal $Ca^{2+}$ transport[20]. Successful confinement of $BaI_2$ within a CNT[21,22] would therefore provide

a powerful system for the alteration of this typical, toxic biodistribution. Essentially, identical methods for molten capillary filling (heating to 811 °C) followed by sonication and washing with water also proved successful for $BaI_2$, thereby supporting the generality of the approach (corroborated by XPS (Supplementary Fig. 7); HAADF-STEM (Fig. 2a; Supplementary Fig. 8); Raman (Supplementary Fig. 5); TGA 8.9% wt (Supplementary Fig. 6)). Pleasingly, when tested as a putative XRF contrast agent (Fig. 2b), the resulting 'Ba + I nanobottle' provided dual signals in the lower spectral window, albeit with a lower relative intensity that is typical of L-lines used here for Ba and I, as opposed to K-lines observed for Pb, as well as the atomic number dependence of fluorescence yield[23]. Further washes confirmed sealing of the CNT end caps by ICP-MS (Supplementary Table 1).

Finally, we tested krypton that, as an unreactive gas, was potentially our most challenging contrast element. Although Kr can potentially condense inside open CNTs at temperatures < 200 K (refs 24,25), there are no reports of irreversible encapsulation. The SWCNTs were sealed into an silica ampoule under approximately atmospheric pressure of Kr, and then heated to 700 °C, before cooling slowly to permit closing of the open nanotube ends[9]. This was followed by work-up under high vacuum to remove any potentially physisorbed gas. Although high-resolution TEM confirmed closed ends (Fig. 2a; Supplementary Fig. 9), the presence of Kr gas was not evident from XPS, HAADF-STEM and TGA (Supplementary Fig. 6) analyses. However, vitally, XRF revealed a strong Kr signal (Fig. 2b) and provided additional confirmation of its unambiguous presence through the measurement of the absorption edge (measured: 14.325 keV; reported[26]: 14.3256 keV; Supplementary Fig. 10). The difficulty in observing Kr using conventional methods for materials is unsurprising, given that gases are on the order of a thousand times less dense than condensed phases. Nonetheless, our first demonstrations of such Kr-encapsulation reveal that despite this lower level of filling, this element also provides a suitable signal for potential use in XRF contrast imaging in one of our identified spectral windows.

**Functionalization of nanobottles for cellular targeting.** Having successfully demonstrated both filling with our chosen contrast elements and the generation of useful XRF signal, we next converted these filled SWCNTs into putative cellular imaging agents through covalent surface attachment of peptides. Representative targeting peptides were selected that would allow us to map the trafficking of these agents at differing depths of penetration into cells: RGD[27] (Arg-Gly-Asp, cell surface adhesion), KDEL[28] (Lys-Asp-Glu-Leu, endoplasmic reticulum (ER) targeting) and two nuclear localization sequences (NLSs) taken from the SV40 virus[29,30] and the human protein IL-1α[31]. A generalized bifunctional spacer was designed and synthesized (Fig. 2c, synthetic details in Supplementary Methods; Supplementary Figs 11–15) to ensure sufficient functional accessibility of peptides to corresponding cellular receptors. This was attached to the side-walls of the filled SWCNTs through the selective reaction of the aniline-moiety terminus of the spacer as its diazonium[32], giving a usefully stable[33] C–C linkage, that leaves the opposite terminus of the linker free for further reaction.

Peptides were constructed through automated solid peptide synthesis using Fmoc (fluorenylmethyloxycarbonyl) chemistry (Supplementary Methods; Supplementary Figs 16–19) and terminated with N-terminal Fmoc-cysteine residues at either the N- or C-terminus, according to the location of the peptides within their native proteins of origin. This allowed a general strategy for chemoselective Michael-type conjugate addition for

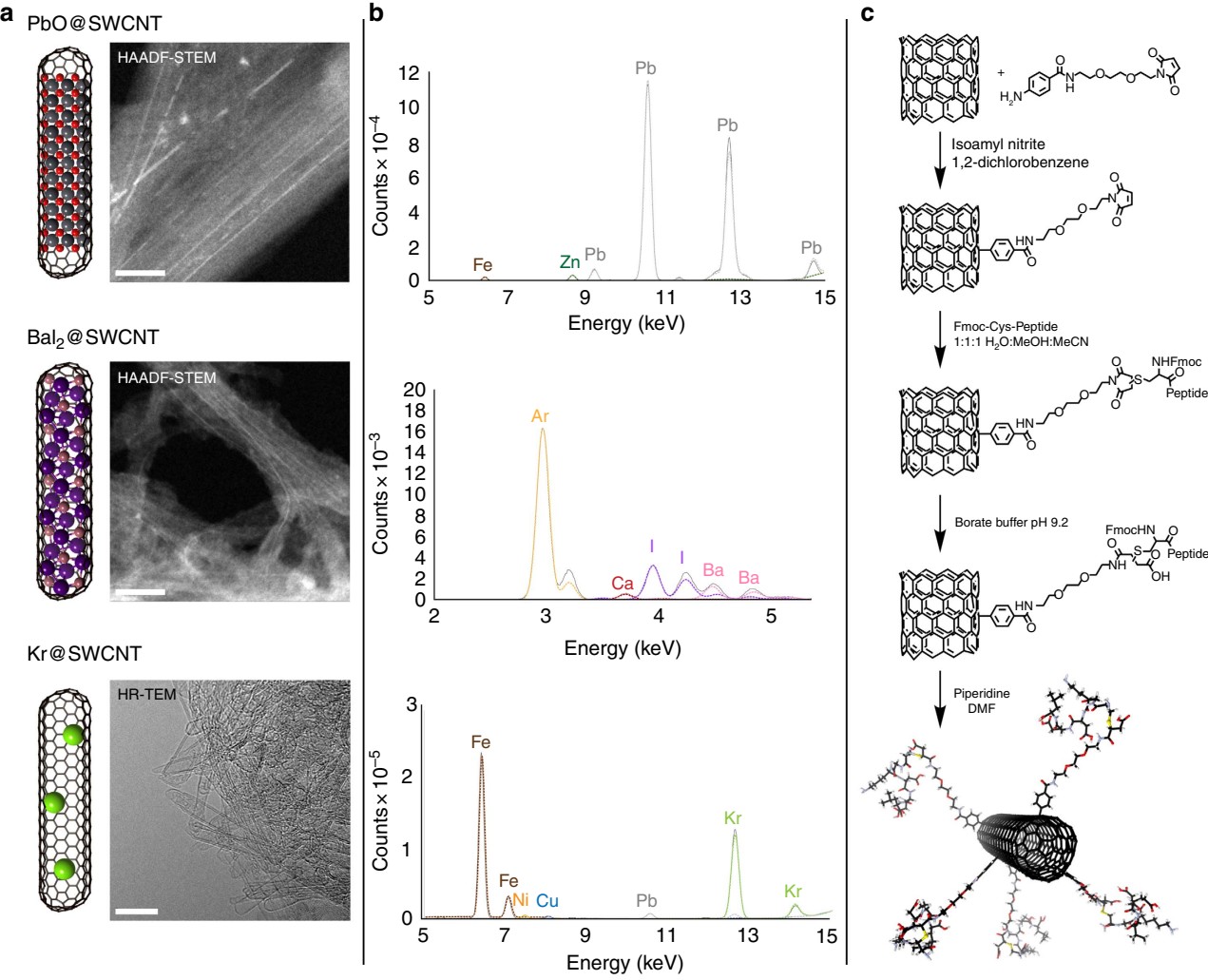

**Figure 2 | Characterization of filled SWCNTs, and functionalization with targeting peptides.** (**a**) HR-TEM characterization of filled SWCNTs (scale bars, 10 nm). HAADF-STEM images show bright lines indicating filling with high-$Z$ species for PbO@SWCNT and BaI$_2$@SWCNT. Although the low density of Kr gas prevented observation of Kr in Kr@SWCNT in this way, the sealing of the 'nanobottle' ends was nonetheless confirmed in bright-field imaging. (**b**) XRF spectra of the filled tubes recorded using synchrotron radiation at excitation energies of 16, 6 and 15 keV for PbO, BaI$_2$ and Kr fillings, respectively, confirm the utility of not only Pb and Ba but also, strikingly, Kr. (**c**) Functionalization of the sealed SWCNTs with peptides was achieved using a novel, bifunctional linker. Further characterization can be found in Supplementary Figs 3–21.

attachment to the maleimide electrophile terminus found on the free end of the SWCNT-bound linkers. Subsequent ring-opening of the resulting succinimide-thioether in basic buffer converted this moiety to a more stable form and hence prevented possible exchange of the resulting conjugate with biological thiols[34,35] during use as cellular probes. Finally, cleavage of the Fmoc group not only generated the final peptide moieties but also allowed ready, precise quantification of the coupling efficiency through UV measurement of the released Fmoc fulvene adduct[36] (which was also corroborated by TGA measurements, see Supplementary Figs 20 and 21); all conjugates showed good loading in the range of 0.2–1.0 µmol mg$^{-1}$ (Table 1; Supplementary Tables 2–4). Peptide attachment in this way noticeably improved dispersion in aqueous media in all cases compared with pristine or linker-functionalized SWCNTs, an important factor for biological use. Biocompatibilities of the peptide-functionalized and filled SWCNTs were tested using 3-(4,5-dimethylthiazol-2-yl)-5-(3-carboxymethoxyphenyl)-2-(4-sulfophenyl)-2H-tetrazolium (MTS) cell viability assay[37] in HeLa (human cervical cancer) cells, and revealed no measurable toxicity at concentrations even up to 50 µg ml$^{-1}$ (Supplementary Fig. 22; Supplementary Discussion), consistent with prior observations for

other SWNT-based systems[38,39]. Importantly, no correlation could be made between cell viability and SWCNT filling material, indicating that the XRF contrast component was isolated from cellular processes by the graphene membrane.

**Table 1 | Yields for peptide decoration of filled SWCNTs according to correlated measurement by both Fmoc numbering and TGA.**

| Decoration | Target | Sequence | Conjugation yield (Fmoc) (µmol mg$^{-1}$) | Conjugation yield (TGA; µmol mg$^{-1}$) |
|---|---|---|---|---|
| Maleimide | NA | NA | NA | 1.01 |
| RGD | Cell adhesion | RGDC | 0.81 | 0.37 |
| KDEL | ER | CGKDEL | 0.62 | 0.30 |
| IL-1α NLS | Nucleus | KVLKKRRC | 0.27 | 0.21 |
| SV40 NLS | Nucleus | PPKKKRKVC | 0.24 | 0.22 |

Italicized amino acids were added to the targeting sequence to permit conjugation.

**Cellular microscopy of XRF nanobottle contrast agents.** With putative XRF contrast agents in hand, we first monitored their dynamic, cellular distribution using a range of complementary, CNT-associated imaging methods. First, dual modes of combined, simultaneous optical and Raman microscopy over a 24 h period (Fig. 3; Supplementary Figs 23–27) in HeLa cells allowed microscopic imaging that mutually correlated well: small aggregates of SWCNTs manifesting as optically opaque regions aligned with high G-band Raman intensity (1,500–1,650 cm$^{-1}$). No G-band intensity or dark aggregates were seen with untreated cells. Different targeting moieties dramatically controlled cellular distribution (Fig. 3). RGD-functionalization resulted in fine lines of SWCNT masses (dark and Raman G-band intense), which reliably followed the contours of the cells, consistent with cell adhesion. KDEL-SWCNTs were found in clumps consistent with intracellular localization on the ER (although it should be noted that it was not possible to observe the ER directly in this method). Co-localization of large, optically dark and Raman G-band intense features with the cell nuclei was conspicuous with both the SV40 and IL-1α NLS peptides; uptake of SV40 NLS-SWCNTs was more voluminous, but the IL-1α NLS-SWCNTs were more closely aligned with the nucleus.

Next, we examined the use of low-voltage HAADF-STEM to further test these observations on the intracellularly targeted SWCNTs at higher magnification. As a result of the low electron density (in the presence of contrast-reducing resin matrix) individual SWCNTs are not visible in cells with this technique. Instead, aggregates of targeted SWCNTs could be readily observed, manifesting as regions of moderately high contrast with a fibrous texture (Fig. 3c). This was unambiguously

confirmed through the use of EELS spectroscopy, which revealed characteristic graphitic carbon peaks at ~292 eV (Supplementary Fig. 28). Strikingly, this aggregate-selective imaging revealed previously unobserved modes of dynamic, cellular trafficking of functionalized SWNTs, for example, IL-1α NLS-functionalized SWCNTs were observed during the process of endosomal uptake at various stages before nuclear localization (Fig. 3d; Supplementary Fig. 29). This excitingly uncovered 'debundling' of SWNTs during internalization into complex endosomal compartments in the cytoplasm. Taken together with the dual optical-Raman microscopy, this suggests that, through such 'debundling', aggregates that are too large to traffic on to the nucleus are 'teased-out' as individual SWNTs through a 'Mikado-like' mechanism before nuclear uptake. Similarly, SWCNTs targeted with the SV40 NLS peptide were also taken up to a significant degree, with aggregates found in numerous endosomal compartments, particularly close to the nucleus (Fig. 3c; Supplementary Fig. 30). Using the KDEL-functionalized SWCNTs, bundles were seen in compartments apparently within the ER consistent with the function of KDEL (Fig. 3b; Supplementary Fig. 31); however, in these cases it was harder to distinguish between SWCNTs and rough ER, and the result cannot be taken as fully conclusive.

With these key aspects of cellular trafficking both confirmed and revealed using the peptide-functionalized 'nanobottles,' we next tested their utility as designed XRF-contrast agents in the 'mapping' of a tissue-like cellular array as a proof of principle. An X-ray excitation energy of 10 keV was sufficient to excite the 'Ba + I nanobottle' contents, as well as the biologically present elements, but for the Kr and Pb nanobottles maps were compiled at 15 and 14 keV (respectively chosen for the elemental tags) with

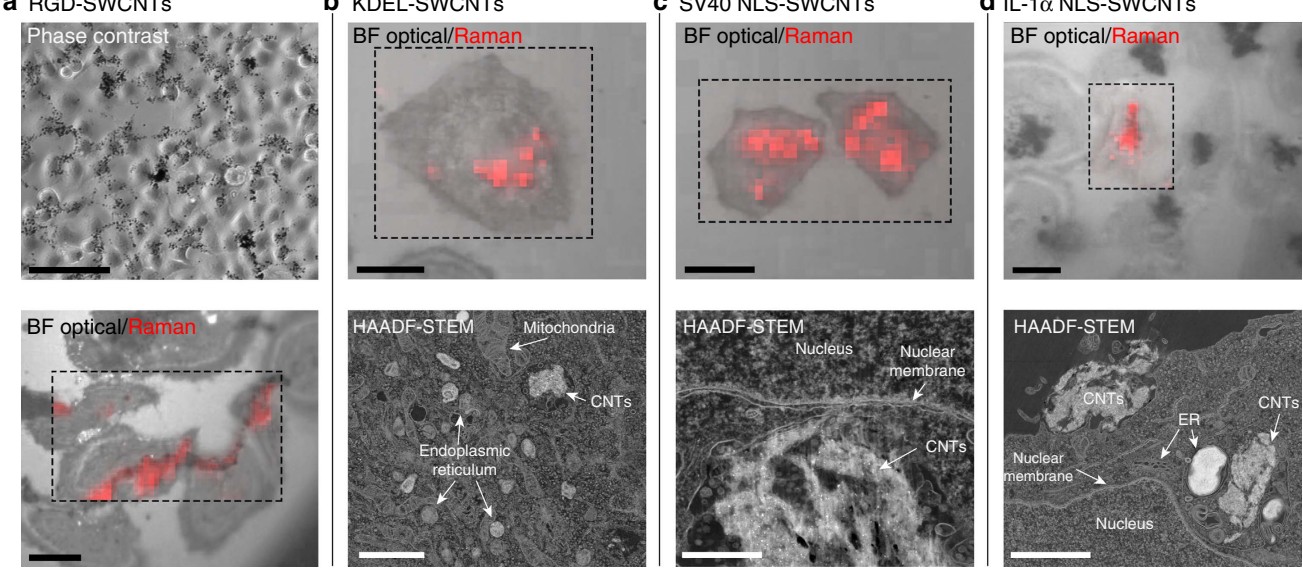

**Figure 3 | Intracellular tracking of peptide-functionalized SWCNT nanobottles.** Maps of the SWCNT Raman G-band (1,500–1,650 cm$^{-1}$) were recorded within the dashed boxes. In the HAADF-STEM images, SWCNTs can be identified by high-contrast regions speckled with even brighter spots (residual iron catalyst particles), a 'fibrous' texture, and a degree of smearing due to the tensile strength of SWCNTs resisting the diamond ultramicrotome knife. (**a**) RGD-functionalized SWNTs interacting with HeLa cells imaged by phase contrast (scale bar, 100 μm), and bright-field microscopy with Raman mapping (scale bar, 20 μm). The SWCNTs are seen to follow the cell outline contours, consistent with the binding of RGD to cell-surface integrins. (**b**) KDEL-functionalized SWCNTs inside HeLa cells imaged by bright-field microscopy with Raman mapping (scale bar, 20 μm) and HAADF-STEM (scale bar, 2 μm). Raman mapping shows SWCNTs within the cells at locations consistent with the ER. HAADF-STEM found SWCNT bundles close to the ER, but could not unambiguously show association. (**c**) SV40 NLS-functionalized SWCNTs inside HeLa cells imaged by bright-field microscopy with Raman mapping (scale bar, 20 μm) and HAADF-STEM (scale bar, 500 nm). Raman mapping showed SWCNTs within the cells, overlapping with the nuclei, and HAADF-STEM revealed interactions with the nuclear membrane, as expected for the association of the SV40 peptide with the nuclear pore. (**d**) IL-1α NLS-functionalized SWCNTs inside HeLa cells imaged by bright-field microscopy with Raman mapping (scale bar, 20 μm) and HAADF-STEM (scale bar, 2 μm). Nuclear localization is clearly visible in the Raman maps. HAADF-STEM uncovered cellular uptake of the SWCNTs and their trafficking towards the nucleus. Further microscopy images can be found in the Supplementary Figs 23–31.

additional scans at 5 keV to provide adequate signal for the lower-Z biologically-abundant elements to provide sufficient contrast (Supplementary Figs 32–37). Maps of untreated cells (Fig. 4a; Supplementary Figs 38 and 39) showed the presence of biotic elements such as P, S, Cl and Zn-ordered diffusely throughout the cells. Of these, the best contrast was obtained for S, and hence the S map was chosen as a marker of the cell outline, which can be used to assess the location of the filled SWCNTs relative to the cell.

In cells that had taken up 'nanobottles', the new contrast element signals could be used to map cellular architecture (Fig. 4b–d) by virtue, in part, of their designed-specific targeting to organelles (vide supra). Small levels of iron impurities are ubiquitous in SWCNTs, and advantageously could be used here as an additional co-localization marker of SWCNT positions[40] (for example, in green channel in Fig. 4b,d). Notably, certain biotic metals (Cu, Zn) were co-localized with the 'nanobottles' (Supplementary Figs 40–52), suggesting mutual influence of Cu- and/or Zn-dependent processes during SWCNT trafficking[41].

Excitingly, the XRF maps of treated cells, enabled through the unique elemental contents of the 'nanobottles' (Fig. 4), revealed modes of uptake and trafficking consistent with those observed by optical, Raman, and electron microscopy imaging detailed above. The location of RGD-functionalized SWCNTs at the cell surface, for example, could be clearly seen using any choice of 'nanobottle' filling material (Fig. 4b–d). Similarly, nuclear location of the SWCNTs was imaged using both the SV40 and IL-1α peptides with all three of these 'nanobottle' contrast systems. The orthogonality of interior contrast agent and exterior peptide targeting is clearly shown. The distributions of barium and iodine in the binary, 'dual-mode' Ba + I 'nanobottle' system were conspicuously well correlated (Fig. 4c), as were all of the contrast elements with Fe, showing that escape was not occurring; this also reduced ambiguities in imaging for improved precision. The power of the 'nanobottle' method for encapsulation was perhaps most markedly evidenced in the fact that it was possible to use Kr as a precisely localized cellular imaging agent (Fig. 4d); this is the first time that this has been achieved to our knowledge.

## Discussion

Scanning XRF microscopy has been used previously to reveal, for example, process-dependent alterations in bioelemental organization[42–44], including disease states[45], and to study the

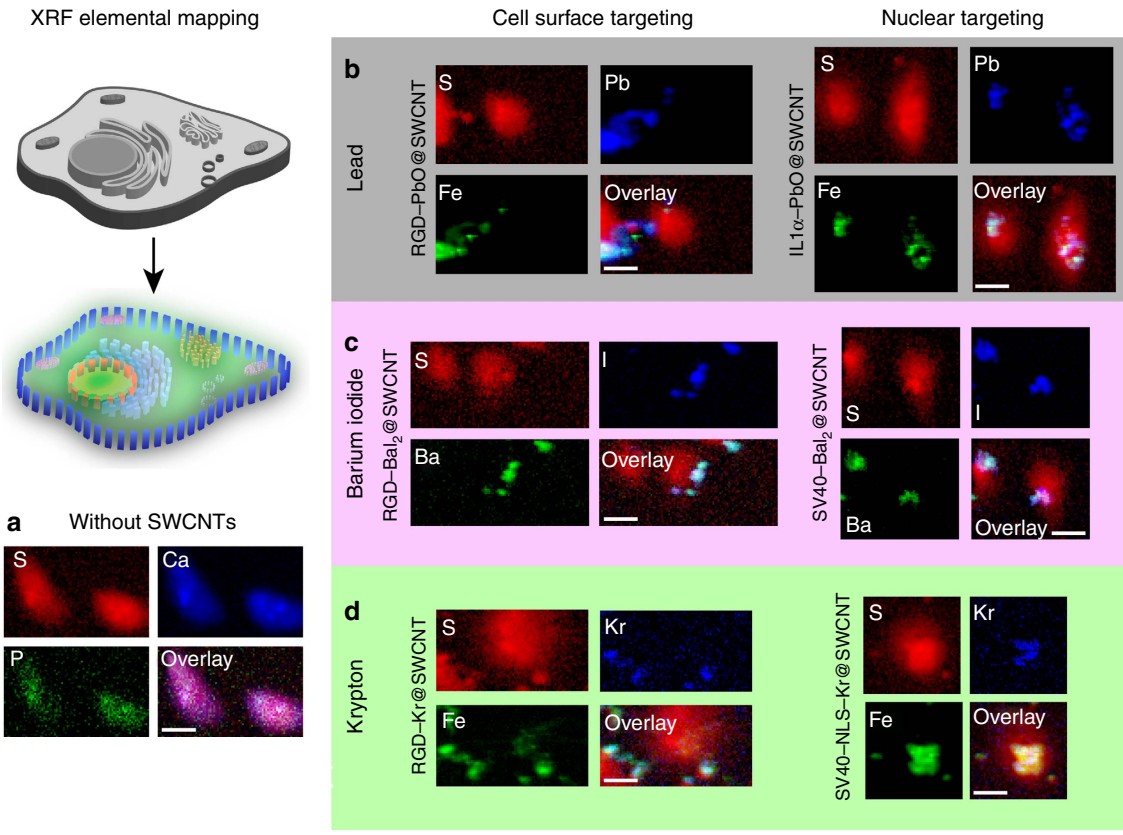

**Figure 4 | Use of filled and decorated SWCNTs as biological XRF contrast agents.** (**a**) Elemental distribution maps of untreated HeLa cells. Three biologically relevant channels are presented to show the cell features. Sulfur was chosen as the best maker of cell morphology. (**b**) XRF maps of cells treated with Pb@SWCNT-based contrast agents. Fe channel is included as a marker of SWCNT location, and correlates with Pb, showing the lack of escape of Pb from the SWCNTs. (**c**) XRF maps of cells treated with BaI₂@SWCNT-based contrast agents. Colocation of Ba and I confirm secure encapsulation of the elements. These elements also correlate with Fe (Supplementary Figs 44–47). (**d**) XRF maps of cells treated with Kr@SWCNT-based contrast agents. Fe channel is included as a marker of SWCNT location, and correlates with Kr, showing the lack of escape of Kr from the SWCNTs. Scale bars (20 μm) in overlay apply to all channels. Maps are plotted on a logarithmic intensity scale; linearly-scaled maps of a wider range of elements ($\geq 8$), and details of deconvolution can be found in the Supplementary Figs 32–52. Synchrotron microprobe arrangements permitted a beam width of 2 μm, with a rounded intensity profile. To maximize the effective resolution obtainable, the cells were seeded sparsely, which results in spread-out morphologies after chemical fixing, meaning that by scanning the beam at 1 μm increments over the sample, the maximum spatial detail can be obtained (at the price of the cells looking 'healthy'). Some peaks corresponding to other, unexpected, elements (for example, Ti, Cr, Ga and La) were also observed across all samples, but since their distribution was uniform (Supplementary Figs 38 and 39), they were included in the fitting calculations to prevent overestimation of the elements of interest, but can be regarded as background noise.

pathways of heavy metal poisons[46,47]. The often diffuse biodistribution of naturally occurring elements in these prior studies means that features, such as organelles and membranes can be poorly defined and ambiguity arises/would arise from use of these elements as labels, thereby hindering interpretation of associated maps. While some elegant studies have made use of xenobiotic elements as contrast agents[48–52], there has been no general method until now that would permit wide-ranging variation of both targeting ligand and XRF tag element.

We believe that the generalizable encapsulation approach shown here will broaden XRF (and potentially other types of) imaging, which is otherwise underused in the biological arena, through the creation of modular and versatile 'XRF contrast agents'. The incarceration of 'difficult' (toxic or gaseous) elements shown here has provided one of the more striking proofs of the impermeability of the atomically-thin graphene carbon membrane[11]. That it is furthermore possible to image the noble gas Kr inside SWCNTs, inside cells, illustrates that such ultra-thin membranes can find useful applications in novel constructs, here as the 'walls' to such 'nanobottles' for imaging. The shielding provided to biological systems by this graphene membrane system from the toxic influence of lead and barium as 'heavy' metals, further illustrates its flexible utility. The elements used as high-contrast tags were chosen primarily according to their unique spectral and hence illustrative potential; given the range of CNT fillings already reported[6,7,53], there is no doubt that many alternative tags are available. This research anticipates the creation of a highly multiplexed library of XRF contrast agents that employ a wide range of elements; the use of binary and higher elemental combinations could even enable a complete cellular 'barcode' system.

It has been argued that the bane of carbon nanotechnology has been its unnecessary application to problems that do not require the unique strengths or properties of carbon nanomaterials[54]. In contrast, here, (i) versatile, hermetic encapsulation via robust, ultra-thin membranes; (ii) diverse modes of imaging and (iii) ready surface-functionalization are essential exploited properties that arise through a unique combination found only in CNTs. The functionalization is achieved chemically and so can be readily extended to other ligands (for example, with an affinity for their organelles[55] or biological markers[56]) in a near-unlimited manner.

Imaging using synchrotron sources spans a huge range of scales. Thus, with microprobe optics, it is now feasible to detect elements in the attogram sensitivity range with pixel sizes <100 nm;[57] while it is also possible to image live, adult humans at the tissue level[58]. 'Imaging' is a word used to encompass all length scales, both microscopic and *in vivo*, but it is rare that a single platform or mode might provide direct access across the board. Correlation here of four distinct modes of 'nanobottle' detection (optical, XRF, Raman and electron microscope) with varying tissue penetration ability excitingly suggests new methods for correlating cellular microscopy with *in vivo* imaging. Additional modes of imaging can also be readily envisaged, for example, through the use of magnetic resonance imaging- or radiochemically active elemental fillings. Although today there are significant hurdles[53] to the use of filled CNT XRF contrast agents in imaging live humans for healthcare (X-ray safety, radiation penetration depths and governmental approval of internal use of CNTs being chief among them), this technology could be used in the near future for highly illustrative *ex vivo* imaging of animal models.

These unique probes have now revealed clear modes of cellular interaction that valuably illuminate the utility of CNTs. As well as the observed lack of toxicity, the effective trafficking of these nanobottles throughout cells (dependent on targeting) has highlighted striking flexibility. In particular, we have observed an effective 'debundling' mechanism that also explains why, although CNT aggregation is fast on a microscopic, physico-chemical level[59], CNTs prove to be effective in their entry into cells, as shown here. These suggest that cellular systems can traffic such CNTs in a wholly effective and dynamic way[60], separating them from each other in a manner similar to intracellular lipid disassembly[61]. Together this suggests that far from being viewed by nature as 'foreign', the decoration of a hydrophobic carbon-based surface with appropriate surface moieties creates conjugates that are well tolerated and even processed, at least in part, naturally. The exploitation of such 'well-accepted nanobottles' could clearly have broad and wide-ranging applications in biology and medicine.

## Methods

Commercially available materials were used without any pre-treatment unless otherwise stated. Dry solvents were obtained by purging with $N_2$ and then passing through an MBraun MPSP-800 column. $H_2O$ was de-ionised and microfiltered using a Milli-Q Millipore machine. All other solvents and commercial grade reagents were used without further purification.

Routine nuclear magnetic resonance (NMR) spectra were recorded on a Bruker AVIIIHD 400 nanobay spectrometer with $^1H$ NMR operating at 400 MHz, proton-decoupled $^{13}C$ at 100 MHz. Routine mass spectra were recorded on a Micromass LCT Premier XE spectrometer and accurate masses determined to four decimal places using Bruker micrOTOF and Micromass GCT spectrometers.

High-resolution TEM and scanning TEM images were acquired on either a JEOL JEM3000F, operating at 300 kV equipped with a JEOL HAADF detector for Z-contrast imaging or a FEI Tecnai G2 F20 microscope operating at 200 kV. Electron energy loss spectroscopy was performed using Gatan Imaging Filter Quantum SE 963 fitted with a $2 \times 2$ k charge-coupled device camera. Low-voltage HAADF-STEM images of cells were acquired at 20 kV on a FEI Magellan XHR 400L FE-SEM equipped with a dedicated STEM detector. Filled CNT samples were deposited from an ethanolic suspension onto lacey carbon copper grids (300 mesh).

Raman spectroscopy was performed using a Horiba Jobin Yvon Raman microscope, exciting using a 632.8 nm laser. Raman mapping was performed in the peripheral lab attached to beamline I18 at the Diamond Light Source synchrotron using a Renishaw InVia Raman microscope using a $\times 50$ objective and exciting at 785 nm.

TGA was performed using a TGA Q5000 IR instrument. About 2 mg of sample were placed into a platinum boat and stabilized for 15 min at room temperature. The sample was then heated under flowing air (20 ml min$^{-1}$) at 10 °C min$^{-1}$ up to 900 °C.

ICP-MS was conducting using a Thermo Finnigan Element 2. Calibrations were obtained using external standards (a series of standards of known trace element concentrations were prepared and analysed to gain a calibration linear, before the measurement of your samples). Dilutions from 10 p.p.m. Merck transition multi element standards traceable to National Institute of Standards and Technology's Standard Reference Materials (NIST SRM's) were used as calibration standards. For a quality check, an external standard was diluted and measured from a dilution of High-Purity Standards 10 p.p.m. ICP-MS-68A and B standard. Sample dilutions were made using a 2% $HNO_3$ solution, prepared using in-house distilled nitric acid and 18.2 MΩ deionized water.

UV/visible spectroscopy was achieved using a SPECTROstar Nano from BMG LABTECH, equipped with a charge-coupled device detector with a spectral range of 220–1,000 nm. The data were analysed with the MARS Data Analysis Software.

XRF and absorption spectroscopy, and XRF mapping were performed at the Diamond Light Source synchrotron, beamline I18. The X-ray beam is focused with a pair of Kirkpatrick Baez mirrors and the energy tuned by means of a Si double-crystal monochromator. XRF maps were analysed using PyMca[62].

**Purification and opening of CNTs.** Chemical vapour deposition Elicarb SWCNTs from Thomas Swan & Co. Ltd. were supplied as dry powder. The as-received material was steam treated to remove carbonaceous impurities and open the ends of the CNTs, following a previously reported protocol[15]. In brief, 400 mg of SWCNTs were finely grounded with an agate mortar and pestle and spread on the centre part inside a silica tube of 5 cm in diameter. The silica tube, opened at both ends, was then placed into the alumina tube of the furnace and purged with argon to allow the complete removal of oxygen. Steam was introduced by flowing the incoming argon through a bubbler with hot water, resulting in a mixture of Ar/steam in contact with the sample. The temperature of the furnace was then increased at 20 °C min$^{-1}$ up to 900 °C, and held at this temperature for 4 h. After this period of time the furnace was cooled down at a rate of 10 °C min$^{-1}$ until room temperature in the presence of Ar/steam. Steam has been reported to remove graphitic particles that coat catalytic nanoparticles[63]. Therefore, the sample was next treated with 6 M HCl to remove any exposed catalytic particles. Finally, the powder was collected by filtration through a 0.2 μm polycarbonate membrane and washed with distilled water until the pH of the filtrate was neutral and dried at

80 °C overnight. Raman $A_D/A_G$ ratio = 4.85. TGA (air) mass loss peak 615 °C, residue remaining = 0.4 wt%.

**PbO@SWCNT.** PbO (10.0 mg) and SWCNTs (10.0 mg) were ground together, and placed in a quartz tube. The tube was evacuated and sealed. The sample was heated to 918 °C for 10 h, then 858 °C for 10 h. This cycle was repeated and then the sample was cooled to room temperature. The ampule was cut open and the solids sonicated with conc. HCl for 1 h before being collected on a polycarbonate membrane (0.2 μm pores), and washed with copious water. Some grey solids were observed, which were determined to be metallic Pb (likely reduced by graphitic impurities), which were separated by addition of ethanol to the vial. The floating carbonaceous material was pipetted from the surface, while the metallic species sank. After high vacuum drying, 9.3 mg of the product was obtained. The filling was also acheived by heating PbO (60 mg) and CNTs (20 mg) at 1,000 °C for 12 h, giving 17.5 mg of product after work-up. Raman $A_D/A_G$ ratio = 7.43. TGA (air) mass loss peak 565 °C, residue remaining = 15.2 wt%.

**BaI₂@SWCNT.** SWCNTs (10 mg) and BaI₂ (30 mg) were ground together in a pestle and mortar, before being placed in a quartz tube. The tube was subjected to high vacuum for an hour before being flame sealed. The sample was heated at 5 °C min⁻¹ to 811 °C, and held at that temperature for 12 h. The mixture was then cooled at 1 °C min⁻¹ to 500 °C before allowing to equilibrate with room temperature. The ampule was opened and water (15 ml) was added to the mixture, which was then sonicated. The CNTs were collected on a polycarbonate membrane (0.2 μm pores), washed with copious water and dried under vacuum, giving 7.5 mg of black powder as the product. Raman $A_D/A_G$ ratio = 4.96. TGA (air) mass loss peaks at 596, 610 °C, residue remaining = 8.9 wt%.

**Kr@SWCNT.** SWCNTs (26 mg) were ground in a pestle and mortar and then transferred to a 15 cm long silica tube. The tube was carefully evacuated and filled with Kr from a balloon four times. The taps were closed, leaving the tube at around atmospheric pressure, and the bottom of the tube was inserted into liquid nitrogen to the maximum extent possible, leaving room for application of the blowtorch at the constriction point without risk of heating the Dewar. Due to the contraction of the gas, the reduced pressure in the tube enabled easy sealing. The sealed tube was then heated to 700 °C at 5 °C min⁻¹, held for 4 h, and then allowed to cool to ambient temperature. The tube was opened and the contents subjected to vacuum for 1 h, giving a total of 15.7 mg of product. Raman $A_D/A_G$ ratio of starting material = 4.85, product = 6.88. TGA (air) mass loss peak 634 °C, residue remaining = 1.1 wt%.

**General method for attachment of maleimide linker to SWCNTs.** Aniline–maleimide linker (0.180 g, 0.518 mmol, 5 equivalents with respect to nanotube carbon atoms) in orthodichlorobenzene (25 ml) for 20 min. Isoamyl nitrite (84 μl, 73 mg, 0.622 mmol) was added and the mixture heated to 100 °C for 24 h. Ethanol (75 ml) was added, and the mixture centrifuged. The supernatant was decanted, and 50 ml EtOH was added, followed by sonication. Centrifugation/supernatant removal was performed two more times with EtOH and two times with water. The SWCNTs were collected on a polycarbonate membrane (0.2 μm pores) and dried under vacuum (51 mg). Raman $A_D/A_G$ ratio of starting material = 4.85, product = 7.53. TGA (air) mass loss peaks 116, 361 °C (organics), 635 °C (CNTs), residue remaining = 0.2 wt%.

**General prep for CNT peptide appendage and Fmoc cleavage.** Peptides (1.39 × 10⁻⁵ mol, 1.5 equiv. with respect to maleimide) were dissolved in 5 ml of water/methanol as necessary for solubility. Maleimide-SWCNTs were added (5 mg) and the mixture was sonicated for five minutes, before stirring overnight. The solid was collected by centrifugation and washed on a polycarbonate membrane (0.2 μm pores) with water. The liquid phases were evaporated to recover leftover peptide. The CNTs were then sonicated in 1 ml pH 9.2 borate buffer and placed in the thermomixer overnight at 37 °C. The CNTs were collected by filtration on a polycarbonate membrane, and washed with water. After vacuum drying, the CNTs were sonicated in 2 ml of 5% piperidine in DMF for 20 min. The mixture was centrifuged and the absorption of the supernatant measured at 301 nm to determine Fmoc concentration. Tenfold volume of water was added to the remaining CNTs, which were then collected by membrane filtration, washed with water and dried under high vacuum.

**Cell studies.** Cells were obtained from an existing HeLa line within the Department of Chemistry at the University of Oxford originally obtained from American Type Culture Collection. The specific line is available on request. When thawing cells from frozen, DMEM media containing 10% FBS and 1% pen/strep solution was used. After the first passage, the pen/step was not used in subsequent growth steps. Antibiotics were used again when the CNTs were added, since this was performed in a fume hood under non-sterile conditions to be as close as possible to the sonicator. CNTs were administered at 1 mg ml⁻¹ after thorough

sonication in tissue culture-grade water. The suspension was added to cells in a ninefold volume of media to give a 100 μg ml⁻¹ final concentration. After incubation with CNTs, a PBS wash was performed before any analysis. Cells were tested for mycoplasma infection on a quarterly basis using Lonza MycoAlert Mycoplasma Detection Kit.

**MTS assay.** The effect of filled and decorated SWCNTs upon cell viability was assessed using CellTiter 96 Aqueous One Solution Cell Proliferation Assay from Promega, following the manufacturer's instructions. In brief, cells were seeded in a 96-well plate at a density of 10,000 per well, and permitted 2 h to adhere. The cells were washed with PBS and the media exchanged for 100 μl of media containing SWCNTs at 50 and 100 μg ml⁻¹ (three replicates per SWCNT sample). The cells were incubated for 24 h before washing and replacement of the media with that containing the MTS reagent. After 2 h incubation, absorption at 490 nm (which is approximately proportional to cell viability) was measured with a well plate reader. Control samples were cells untreated with CNTs, untreated cells killed with 10% DMSO, and 100 μg ml⁻¹ SWCNTs in media without any cells. The latter sample gave the same absorbance as the dead cells, indicating at SWCNT optical absorption has a negligible effect upon the assay.

**Preparation of sapphire discs for XRF/Raman imaging of cells.** Sapphire discs were used as a mount for both XRF and Raman imaging because they give neither XRF or Raman signals that overlap with areas of interest (pre-bought tissue culture treated coverslips have intense Raman signals close to the CNT G-band, as well as elemental impurities). After autoclaving, one side of the slides was made suitable for growth of adherent cells by coating in poly-D-lysine hydrobromide. Following the manufacturer's instructions, 50 ml of tissue culture-grade water was added to poly-D-lysine (5 mg) provided in a septum-capped bottle, followed by vortexing to ensure solution. The discs were laid out in Petri dish and enough of the solution was added to ensure complete coverage. After 5 min, the solution was removed by pipette and the discs were washed three times with tissue culture-grade water (theoretically removing the bromide), after which they were left to dry for 2 h in a sterile environment. The discs were then stored in an air-tight container until use. For cell culture on the discs, they were placed with the treated side up in a 24-well plate. Cells were plated at 30,000 per well in 0.6 ml of media. The cells were given 2–4 h to adhere, before the media was swapped for that containing CNTs well-sonicated (at least 30 min) in media at a concentration of 100 μg ml⁻¹. After incubation for 24 h (typically giving a confluency of 60%), the cells were washed with PBS twice, and fixed as follows. A 2.5% solution of glutaraldyhyde in PBS (1 ml) was added to the cells and they were incubated for 1 h at room temperature. Next, the cells were washed with 2 × PBS and 2 × Milli-Q water, with 2 min permitted for diffusion in each wash. The cells were then dehydrated by incubation in 50% EtOH for 10 min, followed by 70, 90, and 95%, and then 100% EtOH for 20 min three times. After removal of the ethanol, hexamethyldisilazine was added and incubated for exactly 3 min. The hexamethyldisilazine was removed and the samples were placed in a vacuum desiccator overnight to ensure removal of all volatiles.

**Raman imaging.** Raman mapping was performed using a 50 × objective and exciting at 785 nm. Due to problems of drift and sample burnout (due to photothermal excitation of CNTs), the best maps were collected quickly—larger, multicell maps at higher resolutions with full spectra tended to become distorted and/or burnt, wasting many hours and destroying samples. In contrast, by mapping one cell at a time, using 100% laser power (785 nm) but moving quickly with half second collection times, 2 μm resolution and looking at only the region around the G-band (centred on 1,585 cm⁻¹) a single cell could be mapped with sufficient resolution to assess the distribution of CNTs. Cells were no more than ~70% confluent, giving them a more spread-out morphology. The sapphire discs were simply placed on a microscope slide, without further efforts to fix using glue or tape, each of which resulted in drift errors. For data analysis using WiRE 3.4, signal-to-baseline (1,500–1,650 cm⁻¹) quantification was used, since the baseline varied significantly from point to point.

**Preparation of cell samples for STEM imaging.** Samples for TEM were prepared on 13 mm circular glass Thermonox coverslips that had been pre-treated for tissue culture. After cell plating and dosing as above, the cells were fixed with freshly made pre-warmed 2.5% glutaraldehyde, 2% PFA in 0.1M PIPES buffer. Three to five washes of 15–30 min with 0.1 M PIPES containing 50 mM glycine was followed by incubation with 1% osmium tetroxide in PIPES at 4 °C for 1 h. After washing again with Milli-Q water, the samples were incubated with 0.5% uranyl acetate at 4 °C overnight in the dark, followed by further rinsing. The cells were dried using progressively higher ethanol concentrations as described above. Epoxy resin infiltration was achieved by incubation with 3:1 EtOH:Agar100 resin for 1 h, then a 1:1 mixture for 2–3 h and 1:3 for 1 h with rotation and finally with 100% Agar100 overnight and with two further changes in 24 h. The agar pieces were transferred to BEEM capsules filled with 100% Agar100 resin and were polymerized for 1–2 days in the 60 °C oven. Sections of 70 nm were then cut using an ultramicrotome. Four sections were taken from each sample, and 30–40 images were collected from a representative range of areas.

**XRF imaging.** XRF excitation energies were selected after initial testing to optimize visibility of the relevant elements. In particular, the lowest possible energy to give a good emission peak of the xenobiotic element (15 keV for Kr, 14 keV for Pb). A second scan of with an energy of 5 keV was required to observe the low-$Z$ biological elements clearly if the first scan was > 10 keV. At 5 keV, the cells appeared to shrink during the course of the collection, so the lower-energy scan was performed after the higher energy. A beam size of 2.0 (v) × 2.3 (h) μm was scanned over the sample at 1 μm intervals, with an exposure time of 1 s per pixel. After the data collection, the total spectrum of the measured area was fitted to the relevant elements (a process that includes comprehensive deconvolution) using PyMca, and the particular settings saved in a configuration file. This was then checked against individual spots to ensure applicability. The data was then processed to obtain net XRF peak areas for all elements of interest at each map pixel.

Supplementary Methods detail the assessment of SWCNT end-sealing; synthesis of the maleimide linker (with corresponding NMR spectra presented in Supplementary Figs 11–15), peptide synthesis (with high-performance liquid chromatography traces and MS shown in Supplementary Figs 16–19), filling-specific details for maleimide decoration and in-depth XRF parameters; (Supplementary Table 5).

**Data availability.** The authors declare that all the data supporting the findings of this study (post processing) are available within the article and its Supplementary Information files. Unprocessed data relating to this manuscript are available from the corresponding author upon request and XRF, TEM, Raman, XPS, TGA, NMR and microscopy data have been deposited on the Oxford database ORA-data at doi: 10.5287/bodleian:9R2Xr7BAY.

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

## Acknowledgements

C.J.S., R.N.R., S.D.M., M.M., M.K., E.P., B.B., G.T. and B.G.D. were supported by the European Commission FP7-ITN RADDEL (contract number 290023). R.R. was supported by a Felix scholarship. K.W. was supported by a Margaret Pelly scholarship. We are grateful to the Diamond Light Source for the award of beamtime (SP11203/SP12063) and to beamline scientists at I18 for support. Ashley Shepherd at the Surface Analysis Facility of the Department of Chemistry at the University of Oxford collected the XPS data, and Errin Johnson at the Dunn School of Pathology of the University of Oxford prepared biological samples for STEM. We are also grateful to MATGAS for providing access to TGA equipment and Mar Estelles for training and support. We are grateful to Thomas Swan & Co. Ltd. for providing Elicarb SWNT. M.M., M.K. and E.P. have conducted this work as part of the PhD program in Materials Sciences at the Universitat Autònoma de Barcelona. ICN2 acknowledges support from the Severo Ochoa Program (MINECO, Grant SEV-2013-0295). B.G.D. was supported by a Royal Society Wolfson Research Merit Award during the course of this work.

## Author contributions

The project was conceived by B.G.D. and C.J.S. C.J.S. synthesized and characterized the constructs, C.J.S., R.R. and K.W. conducted the tissue culture; C.J.S. performed the associated experiments. CNT starting materials were provided by M.M. and G.T. K.G. provided support and consultation from the beamline. R.N.R. assisted C.J.S. with peptide synthesis. Electron microscopy studies were performed by R.N.R., E.P. and B.B. M.K. conducted TGA measurements. XRF and Raman mapping data were collected by C.J.S., R.N.R., S.D.M., K.W. and R.R. C.J.S., G.T., D.C.A. and B.G.D. analysed the data. All authors read, commented on and approved the manuscript.

## Additional information

**Competing financial interests:** The authors declare no competing financial interests.

