## [Peer Review File · Nature Communications]

Reviewers' comments:

Reviewer #1 (Remarks to the Author):

The authors describe a new XRF imaging approach, where toxic elements acting as XRF emitters are contained inside a sealed carbon nanotube (CNT), which in turn can be attached to molecules to target sub cellular structures for XRF and other novel-mode imaging techniques. I believe it is an excellent idea to use XRF imaging in conjunction with such a targeting method. However, I have two major concerns detailed in the following that need to be addressed on order to justify publication in Nature Communications.

1) What do we learn from the images shown?

In Figure 3 the authors show images of targeted cells using various techniques in an attempt to show the tracking of peptide-functionalized SWCNT nanobottles during targeted cell adhesion, uptake and trafficking. In Figure 4 they show XRF images of filled and decorated SWCNTs as biological XRF contrast agents. Both figures are not clear to me.

The images shown in Figure 3 have vastly differing length scales ranging from 100 microns to 500 nm and it is not made clear or properly described how these images relate to each other, why they were chosen and what one can really learn from them. Are they all taken from one and the same cells? Which figure shows which part? What are the different resolutions?

The images in Figure 4 are also unclear. For example, why is Fe shown in Figure 4b (where it seems to correlate with Pb) and Figure 4d (where it seems to correlate with Kr) but not in Figure 4a (no targeting) or 4b (Barium iodide)? Is Fe a trace element in the targeting agent (Pb is known to have Fe traces) or is it part of the cell? Why is S, Ca and P shown in any of the images? What areas of biological importance are shown in the HeLa cells? How do the images show which areas of the cells have been targeted? How do the images in Figure 4 relate to the images in Figure 3?

The authors state that:

'Excitingly, the XRF-contrast maps, enabled through the unique elemental contents of the 'nanobottles' maps (Fig. 4), revealed modes of uptake and trafficking consistent with those observed by optical, Raman, and electron microscopy imaging detailed above.'

The way Figures 3 and 4 are presented and discussed in the manuscript, I cannot see how this claim is substantiated. In order for the reader to understand what the information content is, these images and their discussion has to be made much clearer.

2) What is the practical importance of this work?

In order to evaluate the practical importance of this research it is critical that the reader is provided with many details currently missing (or not well communicated) in the manuscript:

What is the exposure time per pixel?

How long does a scan take?

What is the flux density and beam size (beam size is mentioned, but buried) used: # of Photons/mm²/s?

What is the corresponding dose?

What is the number of contrast atoms for nanotube, and how does this correspond to the expected XRF signal?

The XRF signal is emitted into 4π sr, so it is critical to know

What is the detector type and solid angle used?

and correspondingly

What is the signal strength for each element during the scans and how does that relate to realistic loading of CNTs?

I think a table that summarizes all that information would be very helpful here. (I understand the supplementary information contains quite a few details regarding the sample preparation, loading quantities of the CNTs and many images using various techniques) but not much about the really critical XRF imaging parameters.

It is not clear to me why these specific elements were chosen. For medical applications on humans they are not really practical as their attenuation lengths are too small. Specifically, the attenuation length ($1/e$) of X-rays in water (real tissue should be very similar) in the energy range of 3.9 - 12.4 keV (the range of XRF signals chosen by the authors) varies exponentially from 0.11 mm to 3.8 mm. This means that any sample that is thicker than a few attenuation lengths of the labeling element's XRF energy cannot be detected efficiently. For a practical application in humans, elements with much higher XRF energies would be required. If the authors intend this method to be more relevant for small-animal studies, they should state so. Generally, the motivation of this work could be made clearer.

Compared to most other imaging techniques, XRF imaging can be very time consuming, as each pixel requires its own measurement. Except for extremely small samples XRF is mostly used to obtain two-dimensional maps for example of a thin section of tissue. The most pressing question is actually of how realistic it is to turn this method into a practical imaging tool. I understand that this might be beyond the scope of this paper, but by providing the details and extrapolating the results to a realistic system, the authors will give the reader a sense of magnitude.

Finally, the following important question, which although this might be beyond the scope of this work, should be addressed or at least mentioned:

How stable are the nanotubes to radiation damage? In other words, what is the risk of creating leaks that might release the toxic agents when exposing them to x-rays at a given dose?

One minor comment:

2-20 keV is not considered soft x-rays in the synchrotron community where soft x-rays range typically between 200 eV and 2 keV.

Reviewer #2 (Remarks to the Author):

This manuscript describes the preparation of novel reagents that can be used for XRF imaging in cells. The XRF imaging is relatively routine. The key innovation is the preparation of single-walled carbon nanotubes that encapsulate different elements, allowing cytotoxic elements to be used in cells without those elements interacting with the cell, and thus allowing one, at least in principle, access to an almost unlimited variety of selectively targeted and selectively labeled probes that can be used in concert with XRF imaging of the native elements. The encapsulation is, to my knowledge, completely novel. In the case of Kr encapsulation, the evidence that Kr is sealed within the nanotube seems pretty compelling -- high vacuum exposure does not remove the Kr. For BaI₂ and for PbO, the evidence seems a bit less compelling. The authors state that there is no cytotoxicity, but I did not find this discussion especially compelling. It seems consistent with the elements being irreversibly encapsulated, but seems, to me, to fall short of proof that they are. Can additional data be included, showing for example that there is no cell mortality under conditions where mortality would be expected, or that there is no loss of soluble salts under conditions where they would otherwise be expected to dissolve?

The importance of this work, it seems to me, lies not so much in these three examples of encapsulation: Ba/I, Pb, and K, but rather in the possibility that this might be generalized to permit barcoding of nanotubes, with hundreds, or perhaps even thousands of unique elemental combinations. Might the authors discuss these prospects a bit?

The manuscript include an unusual number of abbreviations, most of which do not seem to be defined. This makes reading the paper harder than it should be. In addition, I found the language to be a bit flamboyant in places. Careful attention of a copy editor would, in my view, improve the presentation.

Reviewer #3 (Remarks to the Author):

This paper shows the irreversible filling SWCNTs with toxic elements and its application in cellular imaging. The authors have demonstrate a new application of nanotubes as well as XRF imaging using this approach. Overall, this was a well written paper that was well-supported by the data and I recommend publication upon making the following minor revisions:

1. The figure scheme needs to include the sealing of the nanotubes. There is a major leap between having an open ended nanotube and a filled nanotube with close endcaps. The caption says "sealing" but the authors have not shown it in the cartoon. This will confuse the reader unless it is corrected. Also the caption could better describe each step of the process and what the colors mean (green, blue, red).
2. Figure 2: Caption is wrong. It is not the creation of targeted XRF nanobottles, it is rather characterization and targeted functionalization of nanobottles.
3. Figure 3: Caption (a) is wrong. It is only a phase contrast image.

Reviewer #1 (Remarks to the Author):

The authors describe a new XRF imaging approach, where toxic elements acting as XRF emitters are contained inside a sealed carbon nanotube (CNT), which in turn can be attached to molecules to target sub cellular structures for XRF and other novel-mode imaging techniques. I believe it is an excellent idea to use XRF imaging in conjunction with such a targeting method. However, I have two major concerns detailed in the following that need to be addressed on order to justify publication in Nature Communications.

- Thank you for this support – we hope we have addressed these concerns (see below).

1) What do we learn from the images shown?

In Figure 3 the authors show images of targeted cells using various techniques in an attempt to show the tracking of peptide-functionalized SWCNT nanobottles during targeted cell adhesion, uptake and trafficking. In Figure 4 they show XRF images of filled and decorated SWCNTs as biological XRF contrast agents. Both figures are not clear to me.

continued

- The reviewer expressed concern that relevance of the images shown in Figures 3 and 4 was not clear. Figure 3 shows evidence of peptide-mediated targeting of CNTs to specific cellular features. Figure 4 shows how this same strategy can be applied to generate XRF maps of cellular features, using filled and decorated SWCNTs as XRF contrast agents.
- We have revised the text to try and clarify these aspects further.

The images shown in Figure 3 have vastly differing length scales ranging from 100 microns to 500 nm and it is not made clear or properly described how these images relate to each other, why they were chosen and what one can really learn from them.

- The scales and methods of imaging were chosen for intended clarity with respect to the features being observed.
- Specifically, since cell outlines can be readily observed at lower resolution, providing an optical image of this kind allows us to provide evidence for the uniformity of RGD-SWCNT distribution (manifested as black spots), which follows these outlines.
- Next, to show that these ‘black features’ were indeed SWCNTs, Raman mapping was performed. This requires higher magnification in order to be verified: subcellular features are smaller, and require higher magnification, and hence was chose for the Raman mapping images for KDEL, SV40 and, IL-1alpha peptides. The 2D Raman maps are supplemented with TEM imaging of cell sections, with the aim of unambiguously identifying targeting behaviour.
- We hope this is clearer and we have revise the text to clarify our choice of resolution.

Are they all taken from one and the same cells?

- The images all come from the same cell line. In each set of experiments, cells from the same culture (and under the same conditions) were split for each different treatment at the same time. They were then, of course necessarily cultured separately to separate the effects of each treatment and then subsequently treated differently when examined by each different imaging method.
- Thus, for example, the same samples were used for both Raman and XRF mapping of each condition (fixed and dried on sapphire discs), although, since measured on different instruments, images are not necessarily of identical individual cells.
- For TEM, cells were cultured and fixed similarly, but were then infiltrated with epoxy resin and cut into sections using an ultramicrotome.
- Full details have been given in the SI.

Which figure shows which part? What are the different resolutions?

- In Figure 3a, the RGD peptide-modification leads to cell surface adhesion. This can be more easily seen using relatively low magnification in the phase contrast image, illustrating how the CNTs follow the contours of the cells. The RGD peptide thus mediates binding of these CNTs to integrins on the cell surface. Cell outlines can be discerned clearly in low magnification phase contrast. Correlation with the black SWCNT matter is seen to be widespread in this wide field image.

continued

- The Raman mapping overlay (Figure 3a, lower panel) confirms the identity of the CNTs and provides higher magnification. Since the attachment of SWCNTs to cell outlines (a multi-micron-scale phenomenon) has been observed using two microscopies which operate on that scale (the same as that of XRF mapping), it was not expected that any new information would be gleaned using higher magnification techniques.
- In panels b, c, and d of Figure 3, subcellular organelles are targeted using the KDEL (designating endoplasmic reticulum retention) and the nuclear-localisation peptides, which are too small to be resolved using the lower magnification and hence for these panels higher magnification was used. For this reason, the higher magnification Raman mapping overlay, and TEM data were also presented.
- The Raman mapping (Figure 3c,d) shows nuclear targeting for the nuclear-targeting IL-1 α and SV40 peptides, and, correspondingly, evidence of endoplasmic reticulum location using the KDEL (ER-targeting) peptide (Figure 3b), all consistent with mode-of-action in biomolecule alone.
- Electron microscopy (HAADF-STEM) does not provide conclusive evidence due to the similarities between the appearance of the ER and the SWCNTs with this technique.
- However, it did show nuclear-targeted peptide-CNTs being trafficked to the nucleus. Since only bundles of CNTs have sufficient contrast in TEM and these are too large for trafficking through the nuclear pores, it was not possible to see individual tubes inside the nuclei.
- Together, these combined images provide an overall mode-of-action that is consistent with the trafficking properties of the attached peptides.
- Further clarifying details of all of these aspects have now been added to the legends.

The images in Figure 4 are also unclear. For example, why is Fe shown in Figure 4b (where it seems to correlate with Pb) and Figure 4d (where it seems to correlate with Kr) but not in Figure 4a (no targeting) or 4b (Barium iodide)?

- In Figure 4, element mapping was primarily determined by the orthogonality of the red-green-blue channels – in each sample, simply for clarity, three representative elements were mapped in an overlay image.
- Many more element channels were explored and analyzed and details of these are given fully in the SI. For example, in the first control (CNT-free map), 17 elemental channels are presented. These data are important because they show that spectral peaks corresponding to unexpected elements which were required to calculate a full spectral fit (Ti, Cr, etc.) are due to background noise and beamline artefacts rather than cell-specific features.
- In panels (b) and (d), it is the distribution of Pb and Kr which are of interest respectively. The remaining channel was used to map iron, which is usually associated with CNTs due to synthetic impurities, and serves to show that the CNT filling material is still localised with the CNTs (there has been no leakage at the scale available to us in this resolution).
- In panel (c), Fe is not shown because it was important to show both Ba and I.
- However, the directly corresponding Fe map is also shown in the SI (Figs. 41-44 in the revised SI) and confirms collocation of Ba and I with the CNTs.

Is Fe a trace element in the targeting agent (Pb is known to have Fe traces) or is it part of the cell?

- The remaining channel in (b) and (d) was used to map iron, which is usually associated with CNTs due to synthetic impurities, and serves to show that the CNT filling material is still localised with the CNTs (there has been no leakage at the scale available to us in this resolution).

Why is S, Ca and P shown in any of the images?

- In the untreated cells panel (a), S, P, and Ca were shown only because they are relatively biological abundant and illustrate the utility of using the S channel as a cell outline – as a morphological marker.

What areas of biological importance are shown in the HeLa cells? How do the images show which areas of the cells have been targeted? How do the images in Figure 4 relate to the images in Figure 3?

- The images show subcellular targeting through the location of the incarcerated material either around the cell contours (RGD) or the centre of the cell (SV40 and IL-1 α). The relative location of the cell is given by the S map.
- These are therefore directly consistent with the images seen in Fig 3.

The authors state that:

'Excitingly, the XRF-contrast maps, enabled through the unique elemental contents of the 'nanobottles' maps (Fig. 4), revealed modes of uptake and trafficking consistent with those observed by optical, Raman, and electron microscopy imaging detailed above.'

The way Figures 3 and 4 are presented and discussed in the manuscript, I cannot see how this claim is substantiated. In order for the reader to understand what the information content is, these images and their discussion has to be made much clearer.

- We realize that our prior text and SI may not have had sufficient detail and we have now altered the text and legends to provide more explanation and clarity on all aspects.
- We hope that these and the answers above have made this more clear.

2) What is the practical importance of this work?

In order to evaluate the practical importance of this research it is critical that the reader is provided with many details currently missing (or not well communicated) in the manuscript:

- We apologize for these omissions and, as suggested, have corrected these in the manuscript and SI.

What is the exposure time per pixel?

- 1 second per pixel

continued

How long does a scan take?

- We scanned over the sample at 1 μm intervals; the maps were of varying sizes, as shown, and hence took different amounts of time.
- Specific details for each are added to the SI; the average time per map was slightly over one hour.

What is the flux density and beam size (beam size is mentioned, but buried) used: # of Photons/mm²/s? What is the corresponding dose?

- A beam size of 2.0 (v) x 2.3 (h) μm was used.
- The photon flux was ca. 5×10^{11} ph sec⁻¹ for 5 – 10 keV X-rays, and approximately half of that for 14 keV. Flux density was therefore on the order of 10^{11} ph μm^{-2} s⁻¹ for 5 and 10 keV and that for 14keV.
- No radiation damage was seen at scans of 10 keV and higher.

What is the number of contrast atoms for nanotube, and how does this correspond to the expected XRF signal? The XRF signal is emitted into 4 pi sr, so it is critical to know

- This is calculated below - using the sensitivity of the beamline to determine minimum quantities of filling material and using that to estimate numbers of CNTs.

What is the detector type and solid angle used?

- The detector was a 6-element silicon drift detector (from SGX) with a total area of 530 mm² and at distance of 70 mm was subtending ~ 0.1 sr.

and correspondingly

What is the signal strength for each element during the scans and how does that relate to realistic loading of CNTs?

- This is also discussed in more detail below – see discussion of minimum CNT presence for signal.
- Corresponding graphs have also been added to the SI.

I think a table that summarizes all that information would be very helpful here. (I understand the supplementary information contains quite a few details regarding the sample preparation, loading quantities of the CNTs and many images using various techniques) but not much about the really critical XRF imaging parameters.

- Exactly as suggested we have now also added Supplementary Table 5.

continued

It is not clear to me why these specific elements were chosen. For medical applications on humans they are not really practical as their attenuation lengths are too small. Specifically, the attenuation length ($1/e$) of X-rays in water (real tissue should be very similar) in the energy range of 3.9 - 12.4 keV (the range of XRF signals chosen by the authors) varies exponentially from 0.11 mm to 3.8 mm. This means that any sample that is thicker than a few attenuation lengths of the labeling element's XRF energy cannot be detected efficiently. For a practical application in humans, elements with much higher XRF energies would be required. If the authors intend this method to be more relevant for small-animal studies, they should state so. Generally, the motivation of this work could be made clearer.

Compared to most other imaging techniques, XRF imaging can be very time consuming, as each pixel requires its own measurement. Except for extremely small samples XRF is mostly used to obtain two-dimensional maps for example of a thin section of tissue. The most pressing question is actually of how realistic it is to turn this method into a practical imaging tool. I understand that this might be beyond the scope of this paper, but by providing the details and extrapolating the results to a realistic system, the authors will give the reader a sense of magnitude.

- We thank the reviewer for these key ‘overview’ questions which (if we may interpret it as such) focuses on the sensitivity and, in part, the number of contrast atoms that might be needed to provide a signal, as well as potential applications.
- These are excellent questions. With regard to sensitivity, we do not know the exact amount of CNT taken up by the cells – this varies between cells and adaptations of FACS, for example, or other techniques to measure the uptake of a *population* of cells cannot be applied directly to any single cell. Moreover, the absolute sensitivity for a given sample varies in a multiparametric and non-linear fashion with respect to beamline setup and nature of sample; therefore, these values cannot be determined accurately.
varies in a multiparametric and non-linear fashion with respect to beamline setup and nature of sample
- However, that said, using information about the general sensitivity of the beamline, we can perform a rough estimate and have now included this information, prompted by this excellent point. This is now included in the SI, under the XRF experimental details.
- Specifically, synchrotron X-ray microprobes have detection limits in the sub-ppm regime, with the precise values depending, of course, on many factors including how the beamline has been setup and varies for the energy range covered by the elements observed.
- We estimate that for a beam area of $5 \mu\text{m}^2$ scanning a depth of $2 \mu\text{m}$, 1 ppm corresponds to 10^{-17} g – i.e. the attogram regime.
- A conservative limit of detection would be 100 ag, which corresponds to ca. 10^5 heavy metal atoms.
- Given filled SWCNT diameters of 1 nm (as seen by HAADF-STEM – Fig 2, Supplementary Figs 4, 6), SWCNT lengths of 450 nm, and the Pb density of crystalline PbO, the limit of detection is in the region of 20 fully-filled SWCNTs).
- Details of this estimate have been added to the SI.
- To test this estimated limit of detection, we have now included XRF spectra of the relevant peaks taken from points in the maps that illustrate the general quality of signal-to-noise obtained when the cells were dosed at the levels reported here - see Supplementary Figs. 29-34. These seem wholly consistent with this estimate and provide experimental evidence to support the order-of-magnitude of these estimates.
- We thank the Reviewer for this excellent point.

continued

- With regard to potential applications, we see many potential avenues but have tried to highlight here the proof-of-principle; above all we have aimed to initiate the development of new systems using contrast agents for use in an underemployed imaging mode with great potential (we believe).
- We certainly acknowledge the difficulties of imaging live humans using synchrotron XRF in terms of safety, time, and penetration depth. That said, synchrotron imaging of humans has been performed (see for example, Arfelli, F. Synchrotron light and imaging systems for medical radiology. *Nuclear Instruments and Methods in Physics Research A*, 454, 11-25 (2000).) Text to this effect has been added to the discussion section of the manuscript.
- There are some hurdles too for the acceptance of medical use of CNTs, which have not yet been surmounted. Many of these are issues of perception and we have now included reference to a review that seeks to discuss these in more details.
- Thus, to answer the Reviewer's question more specifically: in the short term, we expect this technology to be useful for cellular and *ex vivo* studies, whilst in the long term subcutaneous and potentially deeper imaging of live subjects may be possible as detector sensitivity and nanotube material handling properties improve. Text to this effect has also been added to the discussion section.

Finally, the following important question, which although this might be beyond the scope of this work, should be addressed or at least mentioned:

How stable are the nanotubes to radiation damage? In other words, what is the risk of creating leaks that might release the toxic agents when exposing them to x-rays at a given dose?

- This is an intriguing question. In our hands we did not observe any leakage from the tubes (they could be imaged multiple times with the same result and no further dispersion over the timescale of the experiment). The rate of damage, if any, certainly seems to be less than the rate of damage to the surrounding biological milieu (they are probably the most robust component in such cells!).
- Therefore, we believe that on timescale that any experiments are likely to be conducted they are wholly robust.
- Additionally (as we now indicate prompted by the Reviewer's other useful questions), since the imaging of live subjects is a longer term goal, the unanticipated release of CNT filling during imaging is unlikely to present an immediate concern.

One minor comment:

2-20 keV is not considered soft x-rays in the synchrotron community where soft x-rays range typically between 200 eV and 2 keV.

- We apologize for this error and have now corrected our definition of soft vs. hard X-rays. Many thanks for correcting this.

continued

Reviewer #2 (Remarks to the Author):

This manuscript describes the preparation of novel reagents that can be used for XRF imaging in cells. The XRF imaging is relatively routine. The key innovation is the preparation of single-walled carbon nanotubes that encapsulate different elements, allowing cytotoxic elements to be used in cells without those elements interacting with the cell, and thus allowing one, at least in principle, access to an almost unlimited variety of selectively targeted and selectively labeled probes that can be used in concert with XRF imaging of the native elements. The encapsulation is, to my knowledge, completely novel. In the case of Kr encapsulation, the evidence that Kr is sealed within the nanotube seems pretty compelling -- high vacuum exposure does not remove the Kr. For BaI₂ and for PbO, the evidence seems a bit less compelling. The authors state that there is no cytotoxicity, but I did not find this discussion especially compelling. It seems consistent with the elements being irreversibly encapsulated, but seems, to me, to fall short of proof that they are. Can additional data be included, showing for example that there is no cell mortality under conditions where mortality would be expected, or that there is no loss of soluble salts under conditions where they would otherwise be expected to dissolve?

- As the Reviewer notes, the encapsulation of gaseous Kr and its retention does indeed provide compelling evidence.
- With regard to the specific cases of PbO and BaI₂, similar molten phase filling has been shown to reliably result in closed ends of CNTs [Shao, L.; Tobias, G.; Huh, Y.; Green, M. L. H., *Carbon* **2006**, 44 (13), 2855-2858] consistent with our observations here. However, to further confirm these specific cases, we have performed additional experiments exactly along the lines suggested.
- Thus, we performed washing and analysed the composition of the resultant solution. In all cases, if we treat the tubes exactly as we would prior to a cellular experiment, we see effectively no release in aqueous solution for BaI₂ (< 2 ppb) and little release of PbO (<40 ppb).
- We also have added further details of the toxicity assays that we performed. We realise that these may have been unclear but the use of MTS assay is a standardized method for the assessment of cellular viability. Under these conditions it is clear that there is no measurable toxicity.
- Notably, too, even when we perform exhaustive washing under stringent washing with c HCl (aq) – certainly not cellular conditions ! – even then we see clear retention of ‘cargo’. Thus, should it prove necessary in other or further applications, even such stringent treatment could be applied as an additional quality control measure.
- Details of these additional experiments, further details of the toxicity assays and further discussion have all been added to the SI and the main text.
- We hope this combination of toxicity assays and washing experiments, which we believe provides consistent data on the lack of toxicity, now proves more convincing.

- We should also note that, in fact, as we discover here, the sensitivity of the XRF imaging method as applied through such localized/collected ‘cargoes’ allows us to dose cells with levels well below dangerous levels
- The lethal oral dose of Pb for humans is ca. 450 mg kg⁻¹ of subject [The National Institute for Occupational Safety and Health (NIOSH), Lead compounds (as Pb). <http://www.cdc.gov/niosh/idlh/7439921.html> (accessed 10/05/2016)] and that of Ba ca. 50 mg kg⁻¹ [The National Institute for Occupational Safety and Health (NIOSH), Barium (soluble compounds, as Ba). <http://www.cdc.gov/niosh/idlh/7440393.html> (accessed 10/05/2016)]. The exposure inhalation

continued

limits considered immediately dangerous to life or health (IDLH) are $0.10 \mu\text{g mL}^{-1}$ of atmosphere and $0.25 \mu\text{g mL}^{-1}$.

- Our experiments dosed cells with 0.14 (Pb) and 0.031 (Ba) mg kg^{-1} of cells, which is well under to lethal oral dose, but 14 (Pb) and 3.1 (Ba) $\mu\text{g mL}^{-1}$ of media which is higher than IDLH values (but, of course, such an encapsulated system could not have its contents inhaled since they are ‘caged’)

The importance of this work, it seems to me, lies not so much in these three examples of encapsulation: Ba/I, Pb, and K, but rather in the possibility that this might be generalized to permit barcoding of nanotubes, with hundreds, or perhaps even thousands of unique elemental combinations. Might the authors discuss these prospects a bit?

- This is an excellent point and we thank the reviewer for highlighting the possibility for generalised high-level multiplexed barcoding. This ‘multispectral/multimodal’ (if you will permit the loose phrasing) was indeed a major motivation for us given how much information might be stored. This could indeed allow even notions of barcoding via unique combinations, and we have added this to the outlook discussion.
- We thank the Reviewer for this excellent suggestion.

The manuscript include an unusual number of abbreviations, most of which do not seem to be defined. This makes reading the paper harder than it should be. In addition, I found the language to be a bit flamboyant in places. Careful attention of a copy editor would, in my view, improve the presentation.

- We apologize for this. We have now gone through thoroughly and hope that now all abbreviations are clear in the revised version.
- With regard to the language, we have tried too to moderate this and hope we have gone some way to succeeding. We, of course, welcome any further suggestions to improve clarity.

Reviewer #3 (Remarks to the Author):

This paper shows the irreversible filling SWCNTs with toxic elements and its application in cellular imaging. The authors have demonstrate a new application of nanotubes as well as XRF imaging using this approach. Overall, this was a well written paper that was well-supported by the data and I recommend publication upon making the following minor revisions:

- Thank you for the supportive analysis.

1. The figure scheme needs to include the sealing of the nanotubes. There is a major leap between having an open ended nanotube and a filled nanotube with close endcaps. The caption says "sealing" but the authors have not shown it in the cartoon. This will confuse the reader unless it is corrected. Also the caption could better describe each step of the process and what the colors mean (green, blue, red).

- Exactly as suggested, we have amended the figure to include the sealing of the CNTs. Note that this occurs spontaneously as part of the filling procedure upon cooling, and so although this cannot be considered a separate step we have tried to make it very clear at which stage this takes place.
- We have also, as suggested, amended the legend to make it much clearer what the colours represent (i.e., to illustrate different types of filling).
- Thank you for these suggestions.

2. *Figure 2: Caption is wrong. It is not the creation of targeted XRF nanobottles, it is rather characterization and targeted functionalization of nanobottles.*

- Thank you for spotting this error.
- Exactly as suggested, the caption for Figure 2 has been modified.

3. *Figure 3: Caption (a) is wrong. It is only a phase contrast image.*

- Thank you for highlighting our lack of clarity on this also. In fact, the panel (a) of Figure 3 relates to the two microscopy images but we realise that this was unclear and we have now altered the descriptions accordingly.

Editorial:

- We have adjusted the style of the paper to fit with the suggested guidelines, structural guidelines and enclose the appropriate *Nat. Commun.* checklists.

We look forward to your reply.

Yours sincerely,

Ben Davis, on behalf of all of the authors.

REVIEWERS' COMMENTS:

Reviewer #1 (Remarks to the Author):

The authors have addressed most of my concerns and suggestions and I recommend publication. Some small suggested corrections:

Title:

To the best of my knowledge the word 'multichannel' is not commonly used in the x-ray imaging community and it is also somewhat redundant as it is understood that XRF is sensitive to multiple elements. I would either leave it out or replace it with the word 'multi-element'.

Introduction:

The statement 'Advantageously, XRF is also independent of oxidation state and bonding.' is strictly speaking incorrect. The detailed XRF spectrum, e.g. K-alpha or K-beta can, in fact, show chemical changes. There is a whole science community that uses high resolution x-ray fluorescence (often called emission) spectroscopy of 3d transition metals to characterize their chemical states. It is true that in conventional XRF imaging the detectors don't have the resolution to see these small shifts. Furthermore, XRF imaging can be used at excitation energies close to an absorption edge. Then it can be sensitive to the oxidation state and bonding. This has been used for example for XRF imaging of different sulfur or 3d metal species.

A better statement would be:

'Advantageously, conventional XRF imaging is not sensitive to the oxidation state and bonding of the imaged chemical element.'

I suggest that the authors should include the reference:

Popescu et al Mapping metals in Parkinson's and normal brain using rapid-scanning X-ray fluorescence, Phys. Med. Biol. 54, 651-663 (2009).
and possibly other papers on medical or bio XRF imaging.

Reviewer #2 (Remarks to the Author):

The revised manuscript addresses my concerns.

Reviewer #1 (Remarks to the Author):

The authors have addressed most of my concerns and suggestions and I recommend publication. Some small suggested corrections:

Title:

To the best of my knowledge the word 'multichannel' is not commonly used in the x-ray imaging community and it is also somewhat redundant as it is understood that XRF is sensitive to multiple elements. I would either leave it out or replace it with the word 'multi-element'.

- We thank the Reviewer for the suggestion but we are hoping to capture interest in a community beyond that of just the XRF community. Thus, whilst we would certainly agree that this community understands the potential power of multi-element systems, the broader imaging communities may not. For these communities, 'multichannel' is perhaps better as it conveys the broader intent.

Introduction:

The statement 'Advantageously, XRF is also independent of oxidation state and bonding.' is strictly speaking incorrect. The detailed XRF spectrum, e.g. K-alpha or K-beta can, in fact, show chemical changes. There is a whole science community that uses high resolution x-ray fluorescence (often called emission) spectroscopy of 3d transition metals to characterize their chemical states. It is true that in conventional XRF imaging the detectors don't have the resolution to see these small shifts. Furthermore, XRF imaging can be used at excitation energies close to an absorption edge. Then it can

be sensitive to the oxidation state and bonding. This has been used for example for XRF imaging of different sulfur or 3d metal species.

A better statement would be:

'Advantageously, conventional XRF imaging is not sensitive to the oxidation state and bonding of the imaged chemical element.'

- We have changed this exactly as suggested – thank you.

I suggest that the authors should include the reference:

Popescu et al Mapping metals in Parkinson's and normal brain using rapid-scanning X-ray fluorescence, Phys. Med. Biol. 54, 651-663 (2009). and possibly other papers on medical or bio XRF imaging.

- We have now included the suggested reference.

Reviewer #2 (Remarks to the Author):

The revised manuscript addresses my concerns.

•

Thank

you